# Blood Vessel Geometry Synthesis using Generative Adversarial Networks

**Jelmer M. Wolterink**[*]
Image Sciences Institute
UMC Utrecht
Utrecht, The Netherlands

**Tim Leiner**
Department of Radiology
UMC Utrecht
Utrecht, The Netherlands

**Ivana Išgum**
Image Sciences Institute
UMC Utrecht
Utrecht, The Netherlands

## Abstract

Computationally synthesized blood vessels can be used for training and evaluation of medical image analysis applications. We propose a deep generative model to synthesize blood vessel geometries, with an application to coronary arteries in cardiac CT angiography (CCTA).

In the proposed method, a Wasserstein generative adversarial network (GAN) consisting of a generator and a discriminator network is trained. While the generator tries to synthesize realistic blood vessel geometries, the discriminator tries to distinguish synthesized geometries from those of real blood vessels. Both real and synthesized blood vessel geometries are parametrized as 1D signals based on the central vessel axis. The generator can optionally be provided with an attribute vector to synthesize vessels with particular characteristics.

The GAN was optimized using a reference database with parametrizations of 4,412 real coronary artery geometries extracted from CCTA scans. After training, plausible coronary artery geometries could be synthesized based on random vectors sampled from a latent space. A qualitative analysis showed strong similarities between real and synthesized coronary arteries. A detailed analysis of the latent space showed that the diversity present in coronary artery anatomy was accurately captured by the generator.

Results show that Wasserstein generative adversarial networks can be used to synthesize blood vessel geometries.

## 1 Introduction

The quantitative analysis of medical images showing blood vessels has many important applications. For example, the analysis of coronary CT angiography (CCTA) for the detection of atherosclerotic plaque or stenosis is a clinically valuable tool for diagnosis and prognosis of coronary artery disease [1]. Consequently, quantitative analysis methods for CCTA have long been a topic of interest in medical image analysis. Recently, powerful but data-hungry machine learning methods for CCTA analysis have been proposed [2, 3]. Training of such algorithms requires large and diverse training data sets with accurate ground truths.

One commonly used technique to enlarge the amount of available training data is data augmentation, in which predetermined transformations are applied to the already available training data. However, this is likely to only provide machine learning algorithms with transformations of the available training data. An alternative to data augmentation is the synthesis of completely new data. For coronary artery stenosis detection, such data would consist of geometric models of the coronary artery lumen, along with corresponding CCTA images. To synthesize vessel geometries, model-based methods

---

[*]j.m.wolterink@umcutrecht.nl

1st Conference on Medical Imaging with Deep Learning (MIDL 2018), Amsterdam, The Netherlands.

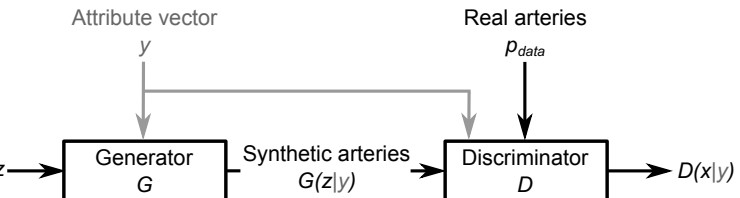

Figure 1: Overview of the proposed method for coronary artery synthesis. The generative adversarial network consists of a generator network $G$ that transforms noise vectors $\mathbf{z}$ from a latent probability distribution $p_z$ into a 4-channel 1D parametrization of a coronary artery. These representations are compared with those of real arteries in a discriminator network $D$, which tries to predict a high score for real arteries and a low score for synthesized arteries. In addition, the generator and discriminator can be conditioned on an attribute vector $\mathbf{y}$ containing information about the real or synthetic samples.

have previously been proposed, in which tube-like vessel structures are synthesized based on a set of heuristics [4]. These vessel structures could then be transformed into corresponding CT images using standard assumptions about CT image formation. Although model-based geometry synthesis allows for a certain amount of control over the synthesized anatomies, the underlying heuristics of such methods may fail to capture the anatomical diversity of real coronary arteries. Hence, in this work we propose *data-driven* blood vessel geometry synthesis in contrast to previously proposed *model-based* approaches. We train a model that aims to accurately capture the data distribution of real coronary arteries. For this we use a generative adversarial network (GAN) ([5], Fig. 1). GANs have previously been used in medical image analysis for tasks such as noise reduction in CT [6], segmentation [7], visual feature attribution [8], cross-modality image synthesis [9]. In terms of vessel structures, GANs have mostly been used to synthesize 2D retinal vessel maps [10].

The goal of this work is to generate plausible 3D blood vessel shapes. In computer vision, volumetric GANs have been used to synthesize meshes or voxelizations of 3D objects such as chairs and cars [11]. A drawback of this approach is that there is no guarantee that the generated object is contiguous; there may be holes or fragments. This is undesirable for vessel shapes. In addition, the size of the generated voxelization may be limited to e.g. $64 \times 64 \times 64$ voxels as in [11]. In this work, we overcome this problem by casting the 3D vessel shape into a 1D parametrization using a set of primitives [12]. Instead of synthesizing a 3D volume, we synthesize a 4-channel 1D sequence representing the vessel central axis. Previously proposed GANs for sequence synthesis have used recurrent neural networks for both the generator and discriminator [13]. In contrast, in our GAN we use convolutional neural networks (CNNs) with large receptive fields, motivated by recent applications of CNNs to long sequences [14, 15].

This work has the following contributions. First, we propose to use an efficient parametrization of blood vessels for generative models. Second, we show how a GAN can be trained to obtain a transformation between a latent space $p_z$ and the space of plausible coronary artery anatomies. Third, we provide a detailed analysis of synthesized vessels and the latent space from which they are sampled.

## 2  Data

### 2.1  Training Data

We use a data set consisting of 4,412 real coronary artery centerlines with radius measurements. These centerlines were semi-automatically extracted from a data set of 50 clinically acquired cardiac CCTA images using a deep-learning based tracking method [16]. Each centerline was extracted based on a manually annotated seed. Seeds were placed approximately equidistantly (10 mm) in the coronary arteries. Because multiple seeds could be placed in the same coronary artery and one centerline was extracted from each seed, the training set can potentially contain overlapping arteries.

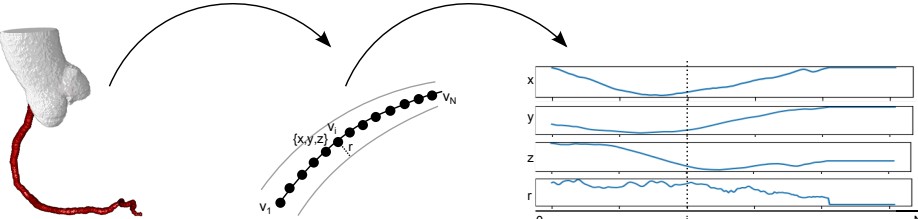

Figure 2: Blood vessels are parametrized according to their central axis, which is represented as a sequence of points $\mathbf{v_1}, \mathbf{v_2}, \ldots, \mathbf{v_N}$, where each point $\mathbf{v_i}$ is a 4-channel vector of $x$, $y$ and $z$ coordinates and a corresponding radius size $r$, assuming a piece-wise tubular vessel.

## 2.2 Blood Vessel Parametrization

While blood vessels are intrinsically three-dimensional, we here simplify the representation of both real and synthesized vessels by using a standard 1D parametrization of the central vessel axis [17, 18]. Each vessel $V$ is parametrized by a central axis or centerline consisting of ordered points: $V = \{\mathbf{v_1}, \mathbf{v_2}, \ldots, \mathbf{v_N}\}$ (Fig. 2). Each point $\mathbf{v_i} \in V$ is characterized by an $x$, $y$ and $z$ coordinate in Euclidean space as well as a vessel radius $r$ (in mm), assuming a piece-wise tubular vessel. This substantially reduces the complexity of the synthesis task. We use the convention that the first point $\mathbf{v_1}$ of a vessel is always located at the coronary ostium.

## 3 Method

We propose to synthesize blood vessel models using a generative adversarial network (Fig. 1). The generative adversarial network consists of a generator network $G$ that can transform a noise vector $\mathbf{z}$ sampled from a distribution $p_z$ into a 4-channel 1D parametrization $G(\mathbf{z})$ of a coronary artery. The discriminator network $D$ compares synthesized coronary artery geometries to real arteries sampled from $p_{data}$ and tries to predict a high score for true arteries and a low score for synthesized arteries. The discriminator and generator can optionally take an attribute vector $\mathbf{y}$ as additional input, which contains characteristics of the coronary artery that is synthesized.

### 3.1 Generative Adversarial Network

The GAN consists of a generator network $G$ which is trained to synthesize blood vessels, and a discriminator network $D$ which is trained to distinguish real from synthesized samples. In the original GAN formulation [5], the generator and discriminator jointly optimize an objective function

$$\min_G \max_D V^{(D)}(D, G) = \mathbb{E}_{\mathbf{x} \sim p_{data}} [\log D(\mathbf{x})] + \mathbb{E}_{\mathbf{z} \sim p_z} [\log (1 - D(G(\mathbf{z})))], \qquad (1)$$

where $\mathbf{x}$ is a sample drawn from the real data distribution $p_{data}$ and $\mathbf{z}$ is a sample drawn from the noise distribution $p_z$. The discriminator tries to maximize this objective function, while the generator tries to minimize it.

We here use a GAN-variant in which the generator tries to minimize the Wasserstein distance between the real data distribution $p_{data}$ and the distribution of synthesized samples, i.e. the amount of work required to transform the synthesized sample distribution into the distribution of real samples [19]. The advantage over the loss function in Eq. 1 is that stronger gradients are provided to the generator by the discriminator at each time step, even when the discriminator can easily distinguish synthetic from real samples. To meet the requirement of a 1-Lipschitz discriminator function as posed in [19], we here enforce a penalty on the gradients between the two distributions [20]. Hence, the full objective becomes

$$\min_G \max_{D \in \mathcal{D}} V^{(D)}(D, G) = \mathbb{E}_{\mathbf{x} \sim p_{data}} [D(\mathbf{x})] - \mathbb{E}_{\mathbf{z} \sim p_z} [D(G(\mathbf{z}))] - \lambda \mathbb{E}_{\hat{\mathbf{x}} \sim p_{\hat{\mathbf{x}}}} [(\|\nabla_{\hat{\mathbf{x}}} D(\hat{\mathbf{x}})\|_2 - 1)^2], \quad (2)$$

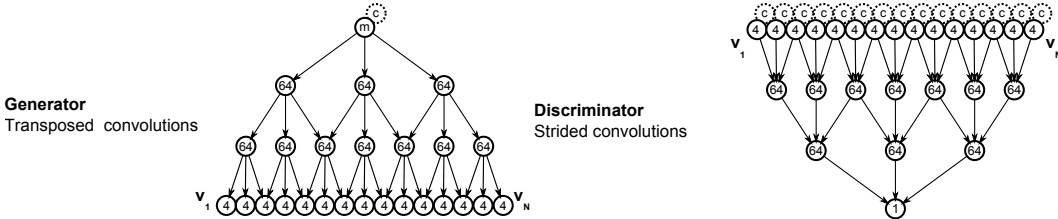

Figure 3: CNN architectures of the generator $G$ and discriminator $D$. The generator uses transposed convolutions in each layer to increase the length of the sequence. The input layer consists of a unit-length sequence with $m$ channels, intermediate layers have 64 channels, and the final layer has four channels which represent $x$, $y$, $z$ and $r$ for each vessel point $\mathbf{v_i}$. The discriminator's architecture mirrors that of the generator, with strided convolutions to compress the sequence into a single scalar prediction. The sequence length $N$ and the number of layers $l$ relate to each other as $N = 2^l - 1$. The generator and discriminator optionally take a $c$-channel attribute vector $\mathbf{y}$ as input.

where $p_{\hat{\mathbf{x}}}$ is the distribution of points along straight lines between pairs of samples in $p_{data}$ and $p_z$, and $\mathcal{D}$ is the set of 1-Lipschitz functions. The gradient penalty is weighted by a factor $\lambda$, which we set to 10.0 as in [20].

## 3.2 Conditional Training

Training the GAN using the objective function in Eq. 2 allows us to sample a wide variety of vessels. However, this provides very little control over the actual appearance and characteristics of the vessels. In some cases, we may be interested in synthesis of only vessels with particular characteristics. For example, given some labels in the training set, we may want to synthesize only left or right coronary arteries, or only vessels with a particular length. We here capture these features in an attribute vector that is provided to both the generator and discriminator network, in the form of an additional input channel [21]. The objective then becomes

$$\min_G \max_{D \in \mathcal{D}} V^{(D)}(D, G) = \mathbb{E}_{x \sim p_{data}}[D(\mathbf{x}|\mathbf{y})] - \mathbb{E}_{\mathbf{z} \sim p_z}[D(G(\mathbf{z}|\mathbf{y})|\mathbf{y})] - \lambda \mathbb{E}_{\hat{\mathbf{x}} \sim p_{\hat{\mathbf{x}}}}[(\|\nabla_{\hat{\mathbf{x}}} D(\hat{\mathbf{x}})\|_2 - 1)^2], \quad (3)$$

where $\mathbf{y}$ is the attribute vector on which $G$ and $D$ are conditioned.

## 3.3 Network architectures

The GAN contains two neural networks: the generator network $G$ and the discriminator or critic network $D$. Both $G$ and $D$ are convolutional neural networks operating on sequences of points, i.e. 1D signals. Fig. 3 shows the CNN architectures used for $G$ and $D$.

Instead of directly synthesizing the $x$, $y$, $z$-coordinates in Euclidean space, we let the generator predict for each point the displacement with respect to the previous point. This simplifies the signal that the generator has to synthesize, as displacements for $x$, $y$ and $z$ are arranged around 0.0. If the length of a training sample is $< N$, we use zero-filling up to $N$ for all four output channels. This facilitates synthesis of vessels with varying lengths. The location of a vessel point $\mathbf{v_i}$ in Euclidean space can be easily retrieved by accumulating all displacements up to point $\mathbf{v_i}$.

The generator $G$ uses transposed, or fractionally-strided, convolutions to rapidly increase the length of a sequence from 1 to $N$. Each convolutional layer consists of a kernel with width 3 and stride 2. Hence, the sequence length $N$ after layer $l$ equals $2^l - 1$. The input to the generator network is an $m$-channel sequence with length 1, with $m$ being the dimensionality of the latent probability distribution $p_z$ from which noise is sampled. In this work $p_z$ is a spherical Gaussian distribution in $m$ dimensions. In the case of conditioning on an attribute vector (Eq. 3), the generator $G$ takes $c$ additional channels, one per attribute. Intermediate layers contain 64 channels. The output layer contains 4 channels for the $x$, $y$ and $z$-displacements, as well as the radius of the vessel along the central axis.

The discriminator mirrors the generator CNN architecture and rapidly reduces the length of the sequence representation from $N$ to 1. The number of convolutional layers $l$ is equal to that in $G$.

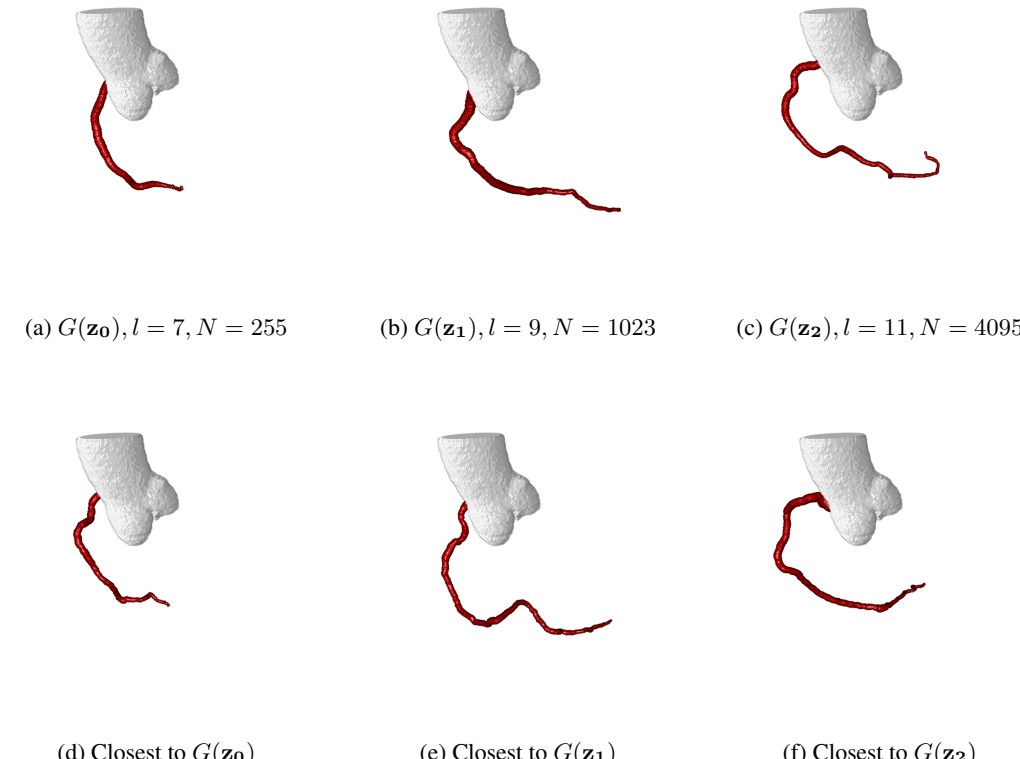

(a) $G(\mathbf{z_0}), l = 7, N = 255$  (b) $G(\mathbf{z_1}), l = 9, N = 1023$  (c) $G(\mathbf{z_2}), l = 11, N = 4095$

(d) Closest to $G(\mathbf{z_0})$  (e) Closest to $G(\mathbf{z_1})$  (f) Closest to $G(\mathbf{z_2})$

Figure 4: The top row shows volume renderings of coronary artery geometries (in red) synthesized at random points $\mathbf{z_0}$, $\mathbf{z_1}$ and $\mathbf{z_2}$ sampled from the distribution $p_z$. The bottom row shows, for each synthesized artery, the closest real coronary artery geometry in the training data set in terms of Hausdorff distance. Ascending aorta (in white) shown for visualization purposes.

Instead of using transposed convolutions, the discriminator CNN uses standard convolutions with width 3 and stride 2. The number of input channels for the discriminator is 4: $x$, $y$, $z$ and $r$. In case of conditioning, the number of input channels is supplemented with one channel per condition. The output consists of a single scalar value, while intermediate layers have 64 channels, as in the generator.

Both the discriminator and the generator use leaky rectified linear units in all layers except for the final layer to stimulate easier gradient flow [22]. In both networks, the final layer uses a linear activation function.

## 4 Experiments and Results

The GAN was trained by alternating updates for $D$ and $G$. Parameters of $D$ were updated according to Eq. 2 using one mini-batch of 64 real samples and one mini-batch of 64 synthetic samples. Parameters of $G$ were updated based on the response provided by $D$ to a mini-batch of 64 synthetic samples. For each update of $G$, we ran five updates of $D$ to make sure that the discriminator was strong enough. All parameters were optimized using two separate Adam optimizers, both with a with learning rate of $\alpha = 0.0001$ [23]. The method was implemented in PyTorch and all experiments were performed on a single NVIDIA Titan Xp GPU. GANs were trained for a total of 200,000 iterations. Training took around 5 hours, while synthesis of 5,000 vessel geometries using a trained generator could be performed in less than a second.

The GAN was trained using the training set of 4,412 real samples described in Sec. 2. To test the ability of the GAN to generate long sequences, we resampled the vessels in the data set to 0.1 mm,

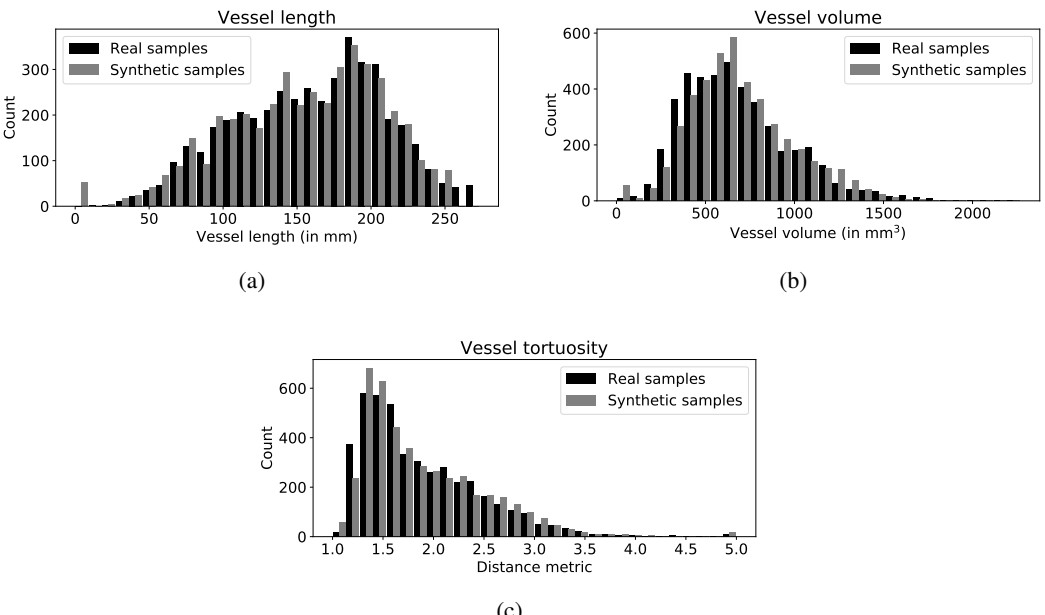

(a)

(b)

(c)

Figure 5: Histograms showing the distribution of vessel length (in mm), volume (in mm$^3$) and tortuosity (using the distance metric [24]) among real and synthesized vessels.

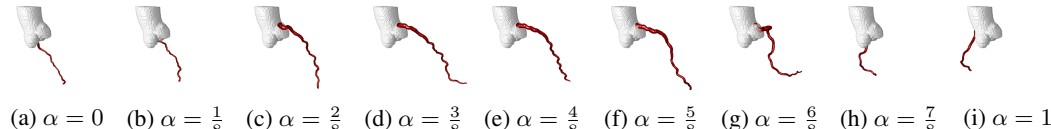

(a) $\alpha = 0$   (b) $\alpha = \frac{1}{8}$   (c) $\alpha = \frac{2}{8}$   (d) $\alpha = \frac{3}{8}$   (e) $\alpha = \frac{4}{8}$   (f) $\alpha = \frac{5}{8}$   (g) $\alpha = \frac{6}{8}$   (h) $\alpha = \frac{7}{8}$   (i) $\alpha = 1$

Figure 6: Vessels obtained by linearly interpolating between two points $\mathbf{z_0}$ and $\mathbf{z_1}$ in the latent space determined by $p_z$. While traversing from $\mathbf{z_0}$ ($\alpha = 0$) to $\mathbf{z_1}$ ($\alpha = 1$), the vessel takes on different orientations, shapes and lengths.

0.25 mm and 1.0 mm. For the lowest resolution, i.e. 1.0 mm, we trained a model with $l = 7$ layers and a maximum sequence length $N = 255$. For resolution 0.25 mm we trained a deeper network at $l = 9$ and $N = 1023$. Finally, for the highest resolution, i.e. 0.1 mm, we trained a model with $l = 11$ layers and a maximum sequence length $N = 4095$. In all cases, the dimensionality of the latent space $p_z$ was $m = 3$. Fig. 4 shows a randomly sampled coronary artery geometry for each of these three models. For each of these generated vessels, we identified the closest real coronary sample based on Hausdorff distance. We find that while there are real samples that are close to the synthesized samples, there are still differences between the real and synthetic samples. This suggests that the synthesized samples do not have an exact matches in the real data distribution, and that the generator learns to synthesize new and unseen samples. More samples are provided online[1].

To assess whether synthesized vessels have similar properties as real vessels, we synthesized 4,412 random vessels and computed their length (in mm), volume (in mm$^3$) and tortuosity (using the distance metric [24]). Fig. 5 shows how these statistics compare to those of real samples in the training distribution, showing strong overlap between real and synthesized samples.

## 4.1 Exploring the Latent Space

The generator $G$ samples coronary artery geometries from the $m$-dimensional probability distribution $p_z$. Different locations in the latent space determined by $p_z$ correspond to different synthesized vessel

---

[1]More images and movies are available online at: `https://tinyurl.com/y99uqk8t`

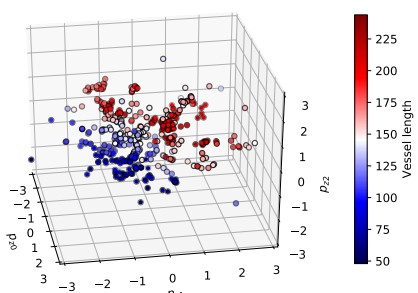
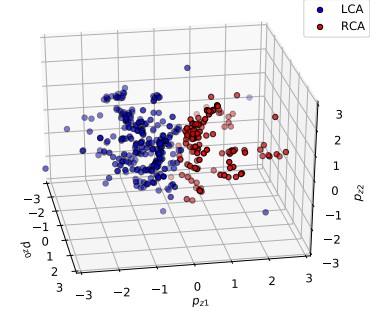

(a) Lengths of unseen vessels mapped to $p_z$        (b) Labels of unseen vessels mapped to $p_z$

Figure 7: Unseen vessel samples mapped to the latent space obtained by GAN training. (a) Marker color indicates length of vessels. (b) Points have been manually labeled as belonging to the left or right coronary artery tree. Marker color indicates whether the vessel belongs to the left coronary artery tree (LCA) or the right coronary artery tree (RCA). Without supervision, the GAN has assigned parts of the latent space to represent either short or long, or left or right coronary arteries.

geometries. To investigate the contents of this latent space, we perform an interpolation in which we linearly interpolate between two input points $\mathbf{z_0}, \mathbf{z_1} \in p_z$. Hence, we obtain several samples $G(\hat{\mathbf{z}})$, where $\hat{\mathbf{z}} = (1 - \alpha)\mathbf{z_0} + \alpha\mathbf{z_1}, 0 \le \alpha \le 1$. Fig. 6 shows the generated vessels when using these interpolated points as input for $G$. This example shows that while traversing the latent space, the generator can consecutively synthesize right coronary arteries (Figs. 6a and 6b), left coronary arteries (Figs. 6c, 6d, 6e, 6f, 6g), and again right coronary arteries (Fig. 6h and 6i). Moreover, different points in the latent space correspond to different vessel length and tortuosity. More samples are provided online[1].

To investigate whether the learned latent space contains structure that generalizes to new data, we trained a GAN using vessels from 45 out of 50 patients. We then mapped the vessels belonging to the remaining 5 patients to the latent space by finding the location $\mathbf{z}$ for which $G(\mathbf{z})$ had the smallest Hausdorff distance to the vessel. Fig. 7a shows how vessels with different lengths are mapped to different locations in the latent space $p_z$. Moreover, we manually labeled vessels as belonging to the left or right coronary artery tree and identified their location in $p_z$. Fig. 7b shows how different areas of the latent space correspond to left and right coronary arteries. However, during training the generator has never been provided with artery labels, and the separation has thus been obtained in a purely unsupervised manner. Such a mapping could potentially be used for automatic labeling of extracted vessels.

## 4.2 Conditioning

While results in the previous section showed that the latent space $p_z$ contains some structure, without knowing exactly what this structure is we can not query the trained GAN for vessels with particular characteristics. To overcome this, we trained a conditional GAN in which the attribute vector $\mathbf{y}$ contained the desired vessel length. The GAN was optimized to generate diverse and plausible samples, while meeting the requirement that the output of the generator matches the desired length. Fig. 8 shows the result of querying the trained generator $G$ with attribute vectors for 50, 100, 150, 200 or 250 mm. The different rows correspond to different locations in latent space, i.e. for each row we have fixed the latent input vector $\mathbf{z}$ and only varied the conditional input vector $\mathbf{y}$ to reflect the desired vessel length. More samples are provided online[1].

The results highlight some of the characteristics of the trained GAN. First of all, samples are quite different for different points in latent space. Shorter vessels (50, 100, 150 mm) are in this case always sampled from the left coronary artery tree. Vessels with length 200 mm are sampled from either the left coronary tree (rows 1 and 2) or the right coronary tree (row 3). The model has learned that very long vessels (250 mm) are more likely to originate at the right coronary ostium. Hence, while the

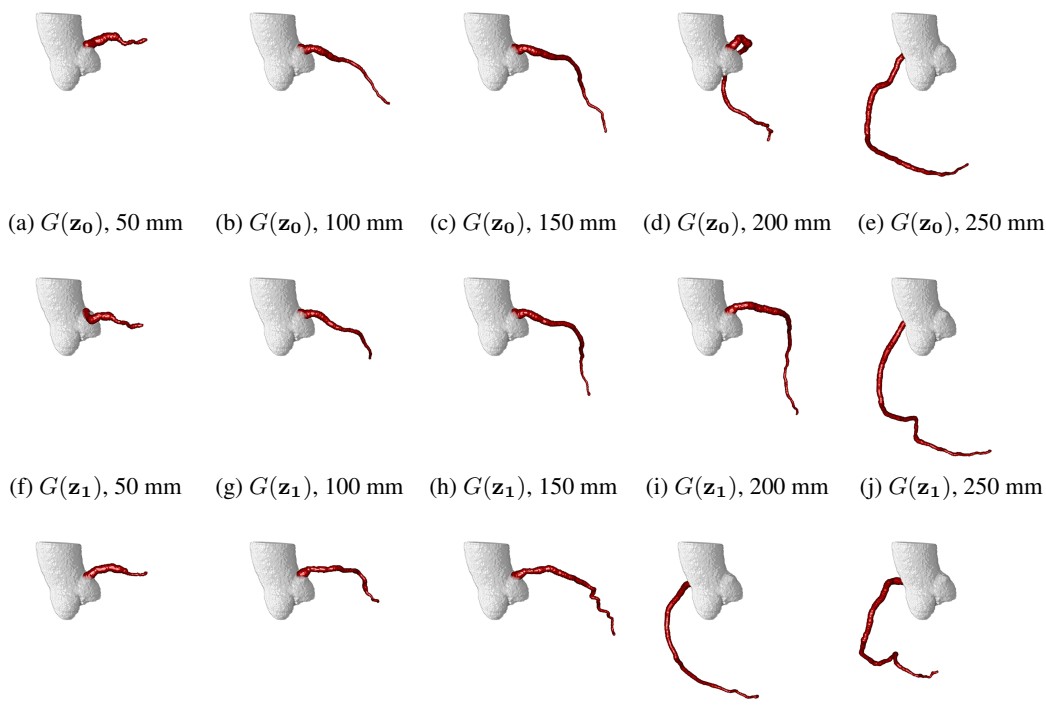

(a) $G(\mathbf{z_0})$, 50 mm    (b) $G(\mathbf{z_0})$, 100 mm    (c) $G(\mathbf{z_0})$, 150 mm    (d) $G(\mathbf{z_0})$, 200 mm    (e) $G(\mathbf{z_0})$, 250 mm

(f) $G(\mathbf{z_1})$, 50 mm    (g) $G(\mathbf{z_1})$, 100 mm    (h) $G(\mathbf{z_1})$, 150 mm    (i) $G(\mathbf{z_1})$, 200 mm    (j) $G(\mathbf{z_1})$, 250 mm

(k) $G(\mathbf{z_2})$, 50 mm    (l) $G(\mathbf{z_2})$, 100 mm    (m) $G(\mathbf{z_2})$, 150 mm    (n) $G(\mathbf{z_2})$, 200 mm    (o) $G(\mathbf{z_2})$, 250 mm

Figure 8: Vessels synthesized with different lengths using a conditional GAN. The rows correspond to different fixed points $\mathbf{z_0}$, $\mathbf{z_1}$, $\mathbf{z_2}$ in the latent space determined by $p_z$. The columns correspond to different conditional inputs $y$ to the generator representing the desired length. Depending on the length and the latent vector, the generator synthesizes left or right coronary arteries.

location in latent space is fixed, the generator prefers to sample a right coronary artery in all three cases.

## 5    Discussion and Conclusion

We have presented a generative method for the synthesis of blood vessel geometries. The generative model learns a low-dimensional latent space that represents the geometry of full coronary arteries. From this low-dimensional latent space a wide range of coronary arteries can be sampled. In addition, our experiments showed how the model can be constrained to only synthesize vessels with particular characteristics.

We found that the synthesized coronary arteries shared statistical properties with the training data set, but that the model also allowed us to synthesize new coronary artery geometries by sampling from the latent space $p_z$. Hence, the model efficiently captures the data present in the training set, yet is able to generate samples that are different from those that is has seen during training. Synthesized geometries could potentially be used to augment training data for discriminative machine learning methods, e.g. those studying flow in blood vessels [25].

The results have shown that the generative method is able to synthesize realistic vessel models using a *data-driven* approach. In previously published work, *model-based* methods have been proposed for vessel synthesis. The proposed method could complement such model-based methods, by introducing realistic variations to the model-based output. In the work by Hamarneh et al., synthesized vessel geometries were transformed into corresponding CT images using assumptions about tissue densities [4]. In future work, we will investigate if the current model can be extended to include such synthesis.

One of the advantages of model-based over data-driven methods is increased control over the synthesized data. However, we found that the GAN organized the latent space into different areas depending on vessel characteristics. Furthermore, we showed that we could control characteristics of the synthesized data using a conditional GAN. We were able to synthesize vessels having different lengths, and to condition the generator network on vessel lengths. In future work, this generative model can be extended to include more vessel characteristics, such as presence and severity of stenosis, vessel location and tortuosity. This requires labeled training samples. In addition, the generator could be encouraged to pass through (user-)indicated key points, thereby allowing more control over the location of the synthesized vessel.

In this work, 3D volumes were synthesized using a set of primitives, namely assuming a locally tubular model for vessels. This substantially simplifies the synthesis task, while yielding contiguous 3D volumes. A similar approach could be used for synthesis of other tubular structures, such as other vessels or airways. In future work, we will extend the set of primitives to include trees and bifurcations, e.g. using graph representations.

In conclusion, we have found that a Wasserstein generative adversarial network can be used to synthesize diverse and realistic blood vessel geometries.

### Acknowledgments

This work is part of the research programme Deep Learning for Medical Image Analysis with project number P15-26, which is partly financed by the Netherlands Organisation for Scientific Research (NWO) and Philips Healthcare.

We gratefully acknowledge the support of NVIDIA Corporation with the donation of the Titan Xp GPU used for this research.

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
