# OpenReview forum: "Blood Vessel Geometry Synthesis using Generative Adversarial Networks"
_MIDL.amsterdam/2018/Conference — MIDL 2018 Poster_

### Review · AnonReviewer3 · 2018-05-07
**Limited applicability, but scientifically solid contribution**

**Rating:** 4
**Confidence:** 3

**Review:**

The paper is well-written and properly summarizes its contributions, namely an efficient parametrization of (coronary) blood vessels for generative models, as well well as experimental results on geometry generation using WGANs. What I like about this paper is that, while it focuses on a narrow topic, it applies solid scientific principles to the chosen topic. The GAN is thoroughly analyzed, design decisions are explained and well-motivated. The biggest drawback of the paper, from my point of view, is applicability / impact: The goal of the presented method is to /generate/ vessel /geometries/. The authors do give some motivations of why this is interesting, for instance when combined with a CCTA imaging simulation for a software phantom, for which, however, one would also have to model surrounding geometries. Within the rather narrow space of vessel geometry synthesis, the contribution is good, though.
The number of centerlines used for training (4,412) is exaggerated (which is explained in the same paragraph), since they describe overlapping geometries from only 50 cases.
The parametrization (4 channels: x, y, z, radius) is based on a "piece-wise tubular" model, but it is not stated which radii are assumed between the support points. (I would assume the final, rendered vessel geometry to be composed of truncated cones and spheres, i.e., using linear interpolation of the radii as well.) The images show voxelized masks and cannot reveal the answer to this question.
Section 3.2 mentions conditional feature vector y, but does not really define "these features". From the experimental results, I would assume that it was only the desired length, but I think that should be stated explicitly.
The paper is 8.5 pages + references, which is roughly half a page more than "strongly suggested". I don't know what to say about this – it still feels quite appropriate, and could be caused by the illustrations. For instance, Fig. 5 could be compressed (one row), and Fig. 8 also has a bit much vertical whitespace. I would not reduce the number of figures, since I think they positively contribute to the paper.
Tiny comments:
- spelling caption Fig. 1: "generator an discriminator"
- misnamed "generator" in "the /discriminator/ network G and the discriminator or critic network D"

**Special Issue:**

Yes

---

### Review · AnonReviewer1 · 2018-05-08
**Nice combination of model-based and data-driven synthesis**

**Rating:** 3
**Confidence:** 3

**Review:**

Authors propose to synthesise realistic blood vessels using Wasserstein GANs. The network is trained with existing data and the generation is empirically analysed.

Pros:
1. The proposed approach is a neat combination of model-based and data-driven approaches for generating examples. Authors’ approach makes sure the generated examples are contiguous and realistic.
2. Authors’ empirical analysis is valuable for understanding how the model behaves.
3. The conditioning during generation is a very nice idea and the empirical analysis shows its value as well.
Cons:
1. Details for the conditioning are missing in the text. It is not clear how the conditioning is fed in the discriminator.
2. The goal for performing synthesis is set as generating additional training or testing examples for image analysis systems. Unfortunately, a complete analysis that shows the value of the proposed method for this task is not evaluated. Therefore, it is uncertain whether the method can indeed generate samples that will in the end yield performance gains in a specific analysis.

**Special Issue:**

No

---

### Review · AnonReviewer2 · 2018-05-09
**This paper presents the reconstruction method of blood vessel in CCTA dataset based on GAN. Dealing with blood vessels in the CCTA is an important problem and authors have solved the problem by approaching it, interestingly. They have expanded to synthesize the blood vessel by condition. The detailed analysis of experiment also has been provided.**

**Rating:** 4
**Confidence:** 2

**Review:**


Quality & Clarity

#1. This paper is well organized, and methodology part was clearly explained.
#2. The description of dataset and experimental results is well written.
#3. The detailed analysis supporting experimental results have been provided.

Originality & Significance

(+) Authors have defined meaningful and interesting problems and solved them with novel idea.
(+) In order to utilize GAN, the parameterization of blood vessel was performed, furthermore, interesting results were obtained through its’ conditioning.

**Special Issue:**

Yes

---

### Comment · ~Bram_van_Ginneken1 · 2018-05-18
**Selection for longlist for special issue Medical Image Analysis**

Dear authors,

Congratulations on your acceptance to MIDL! We have selected your paper on the longlist for the Medical Image Analysis Special Issue. Please read this page:
https://midl.amsterdam/special-issue-in-medical-image-analysis/
Please answer the three questions that are listed on that page about your interest in submitting to the special issue, potential overlap with other publications, and related publications.

You can post your answer here directly below on openreview.net, or mail me directly at bram.vanginneken@radboudumc.nl.

Best regards, Bram

---

### Decision · Program_Chairs · 2018-05-15
**Paper35 Acceptance Decision**

Poster